# Structural basis for modulation of human Na$_V$1.3 by clinical drug and selective antagonist

Xiaojing Li[1,2,3,9], Feng Xu[4,5,6,7,9], Hao Xu[2,8,9], Shuli Zhang[3,4], Yiwei Gao 📧[2,3], Hongwei Zhang[2,3], Yanli Dong 📧[2], Yanchun Zheng[3,4], Bei Yang[2], Jianyuan Sun[3,4,5,6], Xuejun Cai Zhang 📧[2,3], Yan Zhao 📧[2,3,4 ✉] & Daohua Jiang 📧[1 ✉]

Voltage-gated sodium (Na$_V$) channels play fundamental roles in initiating and propagating action potentials. Na$_V$1.3 is involved in numerous physiological processes including neuronal development, hormone secretion and pain perception. Here we report structures of human Na$_V$1.3/β1/β2 in complex with clinically-used drug bulleyaconitine A and selective antagonist ICA121431. Bulleyaconitine A is located around domain I-II fenestration, providing the detailed view of the site-2 neurotoxin binding site. It partially blocks ion path and expands the pore-lining helices, elucidating how the bulleyaconitine A reduces peak amplitude but improves channel open probability. In contrast, ICA121431 preferentially binds to activated domain IV voltage-sensor, consequently strengthens the Ile-Phe-Met motif binding to its receptor site, stabilizes the channel in inactivated state, revealing an allosterically inhibitory mechanism of Na$_V$ channels. Our results provide structural details of distinct small-molecular modulators binding sites, elucidate molecular mechanisms of their action on Na$_V$ channels and pave a way for subtype-selective therapeutic development.

[1] Laboratory of Soft Matter Physics, Institute of Physics, Chinese Academy of Sciences, Beijing 100190, China. [2] National Laboratory of Biomacromolecules, CAS Center for Excellence in Biomacromolecules, Institute of Biophysics, Chinese Academy of Sciences, Beijing 100101, China. [3] College of Life Sciences, University of Chinese Academy of Sciences, Beijing 100049, China. [4] State Key Laboratory of Brain and Cognitive Science, Institute of Biophysics, Chinese Academy of Sciences, 15 Datun Road, Beijing 100101, China. [5] Sino-Danish College, University of Chinese Academy of Sciences, Beijing 100049, China. [6] The Brain Cognition and Brain Disease Institute, Shenzhen Institute of Advanced Technology, Chinese Academy of Sciences (CAS), Shenzhen-Hong Kong Institute of Brain Science-Shenzhen Fundamental Research Institutions, Shenzhen 518055, China. [7] Department of Neuroscience, Faculty of Health and Medical Sciences, University of Copenhagen, 2200 Copenhagen N, Denmark. [8] Division of Life Sciences and Medicine, University of Science and Technology of China, Hefei 230026, China. [9] These authors contributed equally: Xiaojing Li, Feng Xu, Hao Xu. ✉ email: zhaoy@ibp.ac.cn; jiangdh@iphy.ac.cn

Voltage-gated sodium ($Na_V$) channels generate rapid sodium influx to initiate and propagate action potentials in excitable cells. In mammals, at least nine isoforms of $Na_V$s are expressed with tissue specificity[1]. $Na_V1.1$, $Na_V1.2$, $Na_V1.3$ and $Na_V1.6$, encoded by *SCN1A*, *SCN2A*, *SCN3A* and *SCN8A*, are highly expressed in the central nervous system (CNS)[2,3]. Pathogenic variants of these $Na_V$s are associated with neurological disorders including epilepsy, migraine and neuropathic pain[4,5]. Emergent evidence indicates $Na_V1.3$ is important for fetal neuronal development, and mutations in $Na_V1.3$ are related to focal epilepsies and polymicrogyria[6–8]. Furthermore, $Na_V1.3$ was found to be highly re-expressed in injured peripheral sensory neurons, and the resulting hyperexcitability may cause neuropathic pain[9–11]. Therefore, $Na_V1.3$ is an important therapeutic target for anti-epilepsy drugs and analgesics.

Eukaryotic $Na_V$s are structurally closely related with each other because they share high amino acid sequence similarity[12]. $Na_V$s are composed of a large pore-forming α-subunit and regulatory β-subunits. The α-subunit has 24 transmembrane segments organized into four domains (I−IV), each of which contains 6 segments (S1−S6). The first four segments (S1−S4) of each domain form the voltage-sensor domain (VSD), and S5, S6 and the pore-loop between them form the pore module (PM). The intracellular ends of the four pore-lining S6 helices form the activation gate. In addition, the loop connecting *D*III and *D*IV forms the inactivation gate. Four isoforms of β-subunits (β1−β4) are reported to regulate $Na_V$s' kinetics and cell surface expression[13], and they share similar structures with an N-terminal immunoglobulin domain followed by a signal transmembrane segment. Recent advances of structural studies revealed that eukaryotic $Na_V$s share conserved structural features[14–19]. Moreover, multiple natural toxins or clinical-used small-molecule drugs targeting distinct receptor sites in $Na_V$s were revealed at atomic level, which suggest complicated regulatory mechanisms for $Na_V$ functions[20–23]. To date, most local anesthetic or anti-arrhythmic drugs bind to the central pore of $Na_V$s to block sodium conductance. However, the nearly identical pore regions of different $Na_V$s dampen the enthusiasm in finding isoform-selective drug targeting the pore. As a consequence, new drug binding sites on $Na_V$s and isoform-selective drugs are eagerly awaited to minimize potential off-target side effects.

Natural polypeptide toxins or compounds and synthetic drugs modulate the functions of $Na_V$s via binding to at least 6 distinct receptor sites[24]. Site-2 neurotoxins are a group of alkaloids with diverse chemical structures[25,26], which include plant alkaloids of aconitine (from *Aconitum napellus*), veratridine (from *Liliaceae*) and grayanotoxin (from *Ericaceae*), and batrachotoxin first isolated from the skin of the poison dart frog (*Phyllobates aurotaenia*). In particular, bulleyaconitine A (designated as BLA hereafter) extracted from the *Aconitum bulleyanum* plant is an analogue to aconitine. BLA has been prescribed as treatment for chronic pain and rheumatoid arthritis in China since 1985[27,28]. However, the apparent side effects such as causing cardiac arrhythmia and inducing hyperexcitability limit its therapeutic application[29]. Site-2 neurotoxins have been utilized as useful tools to study $Na_V$s function[30]. Biophysical studies revealed that the toxins modulate voltage-dependent activation, inactivation and ion selectivity of $Na_V$s[31–33]. They are known to shift the voltage-dependent activation to more negative potential, thus are considered as activators. Meanwhile, the aconitine analogues reduce peak current amplitude. Mutagenesis studies suggested that site-2 toxins share overlapping but not identical sites inside the central pore[34–36], the detailed binding sites and the underlying modulation mechanisms remain to be fully elucidated. Importantly, synthetic aryl sulfonamide derivates were reported to inhibit $Na_V$s with isoform selectivity at nanomolar potency[37]. ICA-121431 [2,2-diphenyl-N-(4-(N-thiazol-2-ylsulfamoyl) phenyl) acetamide] (designated as ICA hereafter) selectively inhibits $Na_V1.3/Na_V1.1$ with $IC_{50}$ at ~20 nM, which is up to 1000-fold more potent over other isoforms[37]. In addition, electrophysiology studies on point mutated $Na_V$ variants and the crystal structure of a chimeric sodium channel bound aryl sulfonamide antagonist GX936 showed that these antagonists bind to a site inside the $VSD_{IV}$[38]. However, the molecular mechanisms underlying the specific recognition of ICA by $Na_V1.3$ and the inhibition of $Na_V1.3$ remain elusive.

Here, we report the cryo-EM structures of human $Na_V1.3/β1/β2$ in complex with BLA and ICA at 3.3 Å and 3.4 Å, respectively. Together with electrophysiology data, our results demonstrate distinct mechanisms for the modulation of $Na_V1.3$ by the site-2 neurotoxin BLA and the selective antagonist ICA, providing important insights into development of potential isoform-selective drugs.

## Results

**Functional characterization and overall structure of $Na_V1.3/β1/β2$.** We first examined the functional characteristics of human $Na_V1.3/β1/β2$ co-expressed in human embryonic kidney (HEK) 293 cells by whole-cell voltage clamp recording. As illustrated in Fig. 1a, heterologously expressing cells generated robust sodium influx in response to depolarizing pulses and showed fast inactivation within 5 msec. The channel exhibits typical voltage-dependent activation and steady-state fast inactivation with $V_{1/2}$ of $-18.8 \pm 0.4$ mV and $-45.4 \pm 1.2$ mV, respectively (Fig. 1a), consistent with previous reports[8,39,40]. The human $Na_V1.3$ and $β1/β2$ were further co-expressed in HEK293 cells at large scale and purified to homogeneity in detergents (Supplementary Fig. 1). BLA or ICA was added throughout the purification process, respectively. We performed cryo-EM single-particle analysis of the purified $Na_V1.3/β1/β2$ sample (Supplementary Figs. 2–5). The final 3D reconstruction was refined to overall resolution of 3.3 Å and 3.4 Å for $Na_V1.3/β1/β2$-BLA and $Na_V1.3/β1/β2$-ICA, respectively (Supplementary Figs. 2, 4). The cryo-EM density map shows local resolution at 3.0−3.5 Å for both $Na_V1.3$ and β1, indicating stable interaction between them. In contrast, weak and fragmented density for β2 suggests that β2 is mobile or binds weakly to $Na_V1.3$. The high-quality density map allowed us to build reliable models for $Na_V1.3/β1$.

Reminiscent of reported $Na_V$ structures[16–19], the pore-forming α-subunit of $Na_V1.3$ is organized in a domain-swapped manner (Fig. 1b). The overall structure of $Na_V1.3$ is similar to the reported mammalian $Na_V$ structures with root mean square deviation (RMSD) at ~1.5 Å (Supplementary Table 1). The β1 subunit interacts with α-subunit through extensive interactions between the N-terminal immunoglobulin domain and *D*I extracellular loop (ECL), as well as packing of the C-terminal TM helix against *D*III-S2. Despite relative low resolution for the β2 subunit, the disulfide bond between C55 (β2) and C911 (α) that anchors β2 to the α subunit is clearly revealed (Fig. 1b). Superposition of the $Na_V1.3/β1/β2$-BLA and $Na_V1.3/β1/β2$-ICA structures revealed that they are nearly identical (Supplementary Fig. 6a). All four VSDs displayed activated conformation with three or four gating charges above the hydrophobic constriction site (HCS) (Supplementary Fig. 6b). The inactivation gate IFM motif binds tightly to its receptor site (Supplementary Fig. 6a), resulting in a non-conductive intracellular activation gate with an orifice less than 5 Å in diameter (Fig. 1c, Supplementary Fig. 6a). These structural observations confirmed the $Na_V1.3$ was determined in the inactivated state.

To date, thirteen mutations in $Na_V1.3$ identified from patients are linked to human diseases such as focal epilepsy, some of

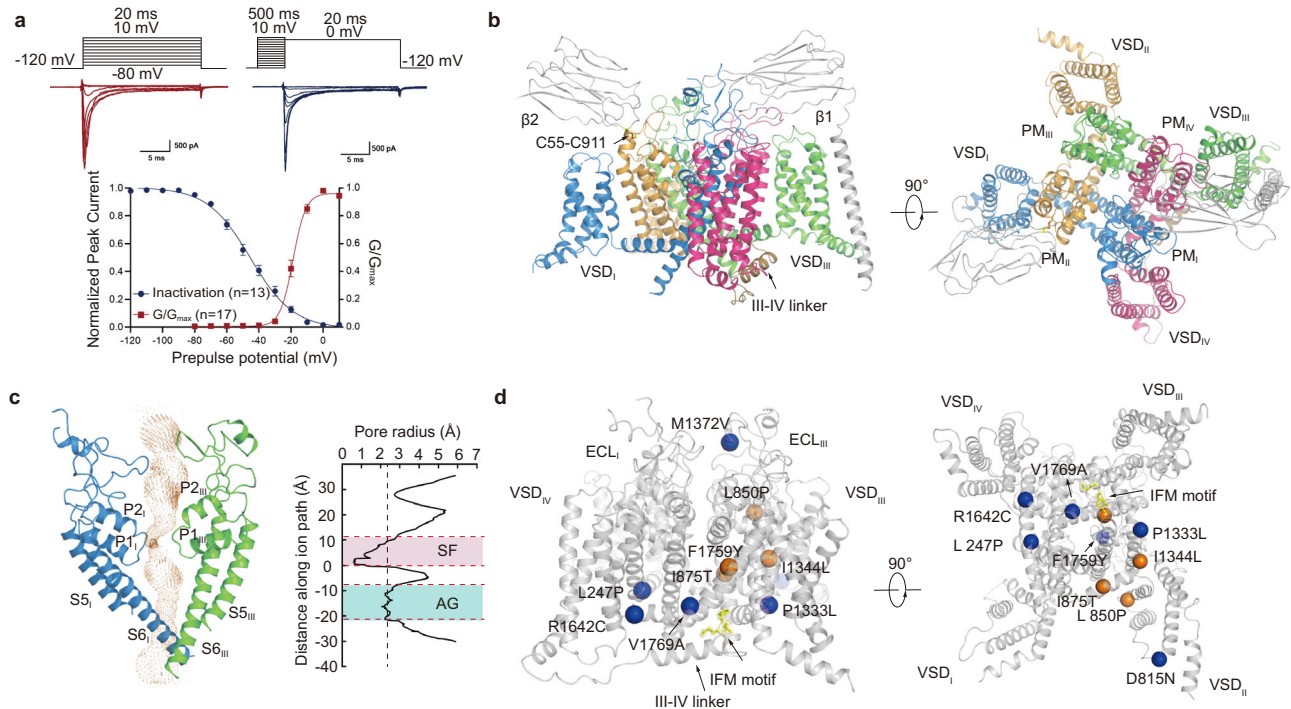

**Fig. 1 Functional characterization and overall structure of human Na$_V$1.3/β1/β2. a** Electrophysiological characterization of Na$_V$1.3. A family of sodium currents conducted by Na$_V$1.3 (Upper panel), a schematic diagram of the recording protocol is presented on top of each current traces respectively. Normalized conductance-voltage (G/V) relationship (Red squares) and steady-state fast inactivation (Blue circles) of Na$_V$1.3 (Lower panel). For measuring G/V curve, Na$_V$1.3/β1/β2 transfected HEK293T cells were measured with 20-ms depolarizing pulses between −80 mV and 10 mV in steps of 10 mV from a holding potential of −120 mV. For measuring steady-state fast inactivation, Na$_V$1.3/β1/β2 transfected HEK293T cells were applied pre-pulse potentials between −120 mV and 10 mV in 10 mV increments for 500-ms followed by a 20-ms test pulse at 0 mV. The Boltzmann distribution has been fitted to each data set, yielding voltage-dependent activation $V_{1/2} = -18.8 \pm 0.4$ mV ($n = 17$) and steady-state fast inactivation $V_{1/2} = -45.4 \pm 1.2$ mV ($n = 13$). **b** Overall structure of human Na$_V$1.3/β1/β2. The β1 and β2 subunits are colored in gray. The α subunit is colored in blue (D$_I$), orange (D$_{II}$), green (D$_{III}$), and magenta (D$_{IV}$). **c** Ion conductance path of Na$_V$1.3 calculated by HOLE. Plot of pore radii for Na$_V$1.3 is shown in the right panel, ligand in the cavity was omitted when calculating pore radius. Two constriction sites are highlighted for selectivity filter (SF) in pink and intracellular activation gate (AG) in light blue. **d** Selected disease-related mutations are mapped on Na$_V$1.3. Blue and orange spheres represent mutations related to focal-epilepsy and polymicrogyria, respectively. Source data are provided as a Source Data file.

which cause severe intellectual disability or polymicrogyria[6–8]. Ten of them are mapped on our Na$_V$1.3 structure (Fig. 1d). L247P is reported as a loss of function mutation[41], whereas I875T and P1333L are gain of function mutations[7]. These mutations are located on the S4-S5 linker helix of each domain, which may cause the related diseases by disrupting the coupling of voltage sensing and gating. F1759Y and V1769A are located at the intracellular end of S6$_{IV}$, which are directly involved in the channel gating. In particular, the gain of function mutant V1769A has been shown to generate large abnormal persistent current at 30% of transient peak current[7], that may be caused by the shorter side chain of V1769A leading to uncompleted closure of the activation gate.

**Mapping the neurotoxin receptor site-2 recognizing BLA.** BLA progressively reduces the peak current amplitude of Na$_V$1.3 in a use-dependent manner (Fig. 2a, b). At 5-Hz frequency, BLA barely inhibited Na$_V$1.3 at the first pulse, however, almost 90% current was blocked after 1000 repetitive pulses (Fig. 2b, right panel; Supplementary Fig. 7a), whereas the control Na$_V$1.3 still generated robust influx after 1000 repetitive pulses (Fig. 2b, left panel). Meanwhile, little inhibition was observed when using a resting-state protocol of holding for 200 sec at −100 mV before test pulse or an inactivation-state protocol including a pre-pulse driving the channel into inactivated state before test pulse (Supplementary Fig. 7b, c). BLA also showed progressive inhibition of

Na$_V$1.3 when applied in intracellular solution (Supplementary Fig. 7d). These data suggest BLA preferably binds to open state Na$_V$1.3. Interestingly, BLA elicits minor and persistent current of <5% of ionic peak current at voltage of −50 mV whereas no current is detected without BLA at the same voltage (Fig. 2c). The efficacy of activation by BLA is much weaker than other site-2 neurotoxins such as batrachotoxin or veratridine[42]. To investigate how BLA modulates Na$_V$1.3, we determined the 3.3 Å cryo-EM structure of Na$_V$1.3/β1/β2 complexed with BLA. Strikingly, unambiguous density located in the central cavity of Na$_V$1.3 close to the fenestration between *D*I and *D*II was observed, which fits well with BLA (Fig. 2d, e, Supplementary Fig. 8a). BLA was sequestered in the cavity by extensive polar and non-polar interactions from P-loops and S6 helices from *D*I and *D*II. The binding was strengthened by five hydrogen bonds between BLA and carbonyl oxygens of T381, C941 and G942, and sidechains of Q382 and N972 (Fig. 2f). The receptor site is defined by M380, T381, V416, I419 and L420 from *D*I, and C941, G942, M968, N972 and L976 from *D*II (Fig. 2f). These interactions make BLA bound so tightly to the receptor site that the binding is nearly irreversible (Supplementary Fig. 7e). Sequence alignment around the receptor site among the nine Na$_V$ isoforms showed most of the key residues are identical or highly conserved, which explains the weak selectivity of the BLA (Supplementary Fig. 7g).

Our complex structure revealed a detailed BLA binding site formed by pore modules of *D*I and *D*II. Consistent with this structural observation, previous studies showed that mutations at

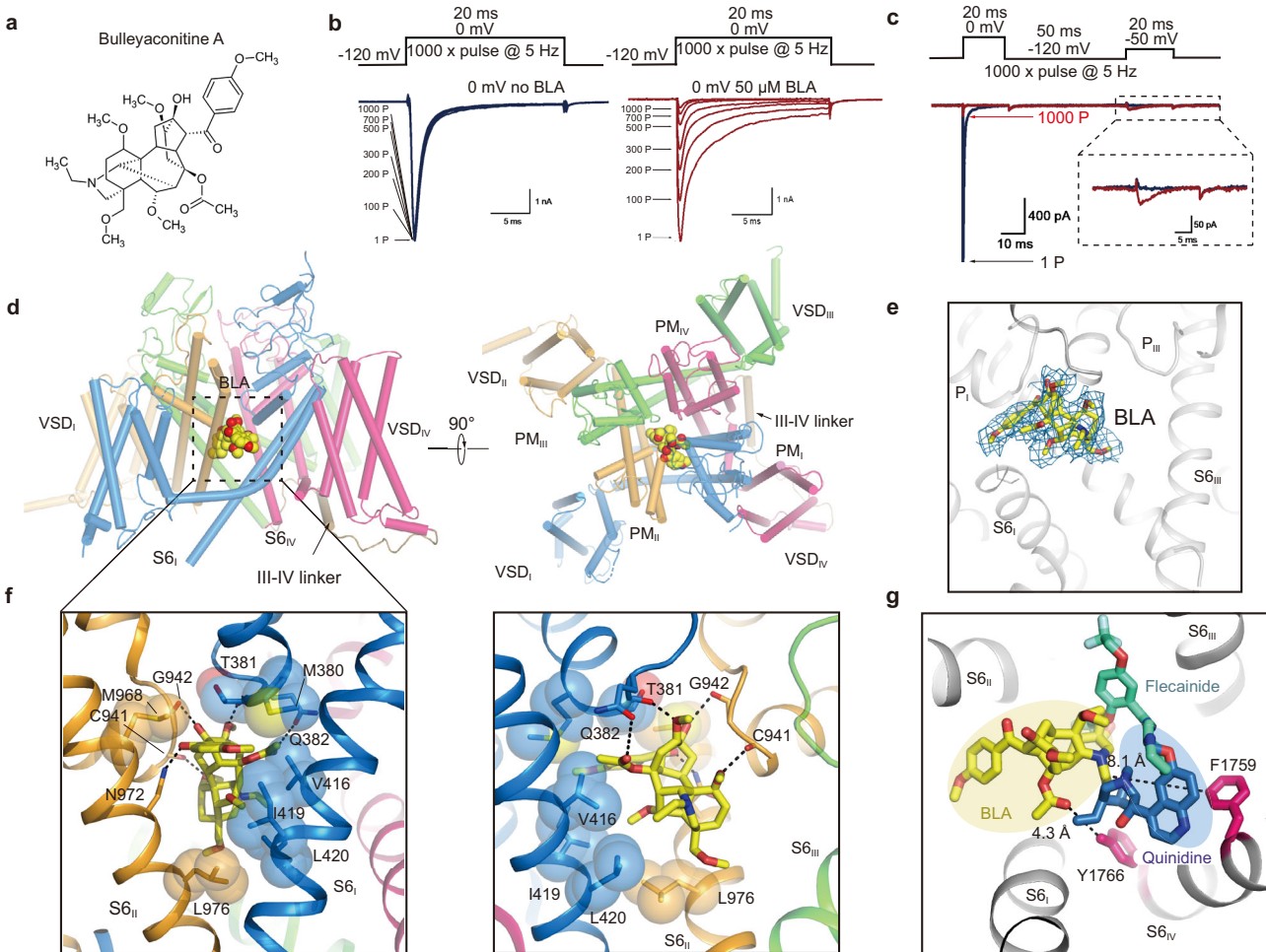

**Fig. 2 The binding site of BLA in Na$_V$1.3. a** The chemical structure of BLA. **b** BLA reduces peak current of Na$_V$1.3 in a use-dependent manner. Na$_V$1.3 transfected HEK293T cells were measured using a recording protocol of 1000 repetitive test pulses at 0 mV from holding potential (HP) at −120 mV at 5-Hz frequency without BLA (left panel), the traces showed robust and stable sodium influx after 1000 pulses. Transfected cells in bath solution containing 50 μM BLA were measured using the same recording protocol and the current traces showed progressive inhibition of Na$_V$1.3 (right panel). Similar data were acquired from 6 cells in the absence or presence of BLA. **c** BLA elicits weak activation of Na$_V$1.3 after binding. Two-pulse protocol, which is composed of a first 20-ms test pulse at 0 mV, backing to −120 mV for 50-ms followed by a second 20-ms test pulse at −50 mV, was used to elicit the activation of Na$_V$1.3 by 50 μM BLA for 1000 repetitive pulses at 5-Hz frequency. The current trace of the first pulse (1 P) and the 1000 pulse (1000 P) are colored in black and purple, respectively. Similar data were acquired from 6 cells. **d** BLA binding site in Na$_V$1.3. The complex structure is shown in side view (left panel) and top-down (right panel) view with BLA depicted in sphere models. The black dashed square indicates the area to be shown in panel (**f**). **e** The density at 3σ is shown in blue mesh for BLA, which is depicted in sticks. **f** Detailed binding site for BLA showing interactions between BLA and Na$_V$1.3. The side chains of key residues are shown in sticks. Black dashed lines represent hydrogen bonds. **g** Comparison of the BLA binding site in Na$_V$1.3 with the flecainide and quinidine binding sites in Na$_V$1.5. Black dashed lines indicate the closest distances from BLA to F1759 and Y1766.

key residues on S6 of *D*I and *D*II result in toxin-insensitive phenotypes[43]. For example, mutations of I433 and L437 (equivalent to V416 and L420 in Na$_V$1.3) on *D*I-S6 and N784 and L788 (equivalent to N972 and L976 in Na$_V$1.3) on *D*II-S6 of rat Na$_V$1.4 diminished the efficacy of site-2 neurotoxins such as batrachotoxin, grayanotoxin or veratridine[34–36,44]. In our structure, V416, L420, N972 and L976 directly engage BLA, which suggests that these interactions are conserved for the site-2 neurotoxins. No direct interaction from *D*III or *D*IV was observed for BLA binding in our structure. However, residues from S6 helices of *D*III and *D*IV were also reported to be involved in batrachotoxin binding[43,45]. These data suggest that site-2 neurotoxins share a common site but not identical binding modes, presumably owing to their structural diversity (Supplementary Fig. 7f). Interestingly, local anesthetic and anti-arrhythmic drugs are physical pore-blockers that bind in the central cavity close to S6$_{IV}$[17,21,46]. The key residues F1759 and

Y1766 are away from BLA at distances of 8.1 Å and 4.3 Å, respectively, which indicates that the receptor site for site-2 neurotoxins is distinct from that for local anesthetic and anti-arrhythmic drugs (Fig. 2g).

**Mechanism for activation and inhibition of BLA on Na$_V$1.3.** Aconitine and BLA preferably bind to the open state of the channel and reduce peak amplitude in use-dependent manner. Once bound, however, they increase its open probability[47]. These ostensibly contradictory effects can be explained by our Na$_V$1.3-BLA complex structure. The functionally closed activation gate of the complex structure shows that the gate is too small for BLA to access the central cavity (Figs. 1c and 3a, b). Comparing to the open-state structure of Na$_V$Ab[48], the open activation gate appears to be large enough for BLA to pass through (Fig. 3c), although one possibility cannot be ruled out that BLA accesses the receptor site through enlarged fenestrations during state transition

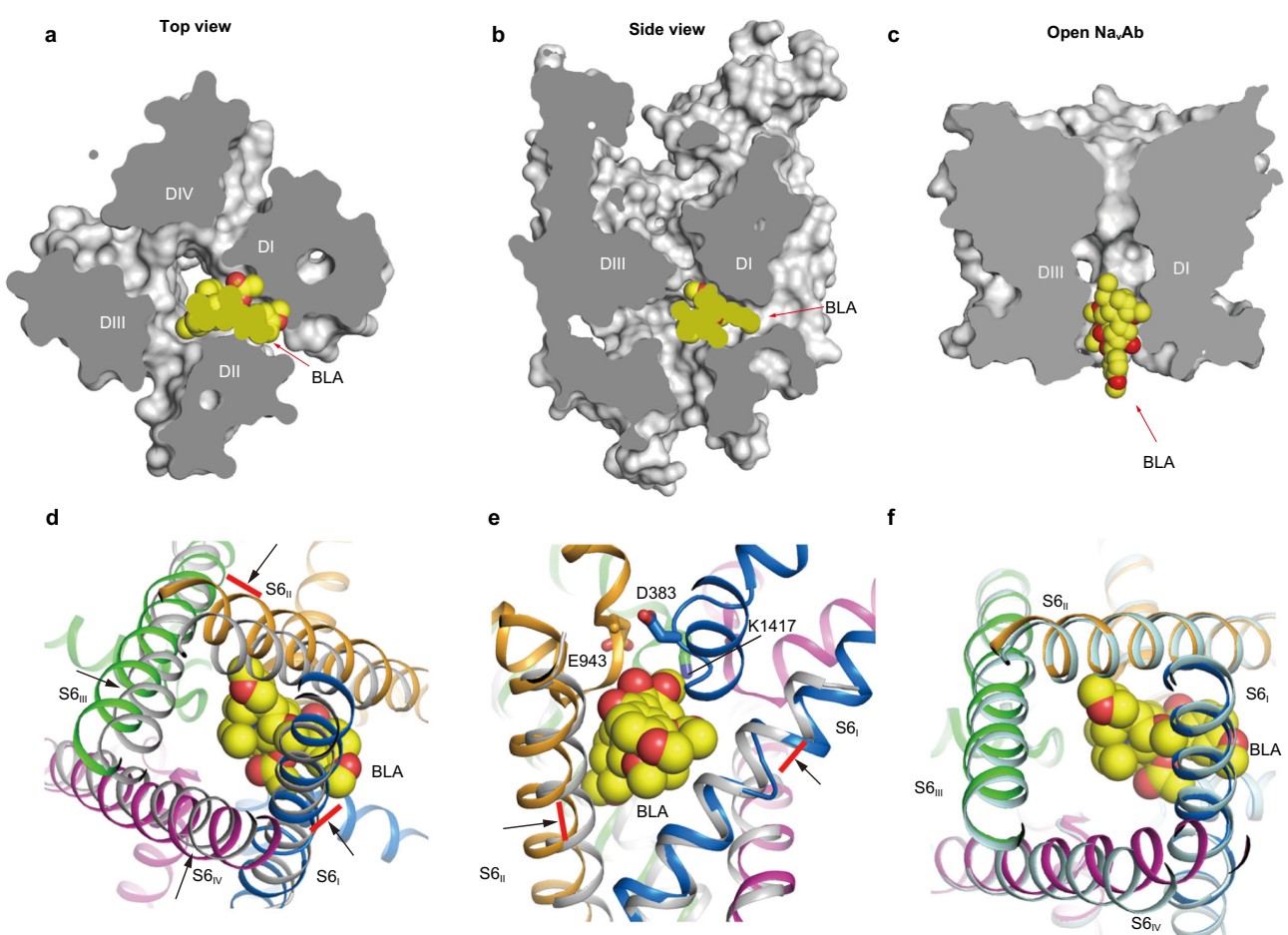

**Fig. 3 Structure basis for modulation of Na$_V$1.3 by BLA. a, b** Cut-open sliced views of the Na$_V$1.3 pore module shown in surface from top-down and side, respectively. **c** Space-filling of the open activation gate of Na$_V$Ab (PDB code: 5VB8) with BLA shown in spheres. **d, e** Binding of BLA physically blocks the contraction of Na$_V$1.3 activation gate compared to the closed gate of resting state Na$_V$Ab (PDB code: 6P6W, colored in gray). Black arrows indicate shift of each S6 when transition to the closed resting state, and red bars indicate where the shifts are blocked. BLA is shown in a sphere model, and key residues for the selectivity filter are shown in sticks. **f** Structural comparison of the activation gate of Na$_V$1.3 with the open state of Na$_V$Ab (PDB code: 5VB8, colored in light blue).

(Fig. 3a). Once integrating into the receptor site formed by the X-crossing S6 helices and P-loops of $D$I and $D$II, BLA physically prevents the shift of the two S6 helices back to the resting state compared to the resting state Na$_V$Ab[49] (Fig. 3d, e). Superposition of the activation gate between BLA-stabilized Na$_V$1.3 and open Na$_V$Ab shows that these two intracellular gates are fairly superimposable (Fig. 3f). We therefore postulate that BLA binding stabilizes the open conformation of S6 of $D$I and $D$II at least, which may improve the open probability of the channel. On the other hand, BLA sits underneath the DEKA locus (Fig. 3e), the signature motif that determines sodium selectivity; in addition, three of the five hydrogen-bonds with BLA are attributed to the P-loops of $D$I and $D$II (Fig. 2f), implying a mechanism by which BLA analogue aconitine alters ion selectivity of Na$_V$1.4[50]. Furthermore, the bulky BLA partially blocks the ion path, which would surely be an obstacle for conducting hydrated-Na$^+$ (Fig. 3a, b). Our structure demonstrates that BLA binds to the unreported receptor site and stabilizes the open conformation of S6 of $D$I and $D$II, meanwhile, it closely interacts with P-loops and partially blocks the ion path resulting in change of ion selectivity and reduction of peak current.

**Detailed receptor site in VSD$_{IV}$ determines selectivity of ICA.**
We next examined the effect of the selective ICA on Na$_V$1.3. ICA exhibited negligible inhibition on Na$_V$1.3 when test pulse was applied at holding potential of −120 mV (Fig. 4a, b). In contrast, ICA almost completely inhibited the inward current of Na$_V$1.3 when the test pulse was proceeded by a protocol to drive the channels into inactivated state (Fig. 4b, right panel). ICA showed strong inactivated state inhibition with IC$_{50}$ of 95.5 ± 9.3 nM (Fig. 4c). The state-dependent inhibition is consistent with previous report[37].

To investigate the selectivity and inhibition mechanism of ICA, we determined the cryo-EM structure of Na$_V$1.3/β1/β2-ICA at 3.4-Å resolution (Fig. 4d, Supplementary Figs. 4–5). Notably, a strong piece of density was found inside the extracellular aqueous cleft of VSD$_{IV}$, which is absent in the map of Na$_V$1.3/β1/β2-BLA and agrees well with the ICA molecule (Fig. 4e, Supplementary Fig. 8b). The complex structure clearly revealed a detailed receptor site for ICA. The small sidechains of G1603 on S3 helix (G1603$^{S3}$) and A1626$^{S4}$ create room to accommodate the thiazole headgroup (Fig. 4f). The anionic thiazole directly interacts with three of six gating charges (i.e., R2−R4) through electrostatic interactions (Fig. 4f). The middle benzene ring contacts L1563$^{S2}$ and M1604$^{S3}$ via van der Waals interactions. The protruding diphenyl tail of ICA outside of the binding cleft is further stabilized by R1560$^{S2}$ and F1605$^{S3}$. Interestingly, a possible lipid molecule helps to stabilize the hydrophobic tail at a distance of ~4 Å, suggesting that lipid may be involved in this drug-channel interaction (Fig. 4e).

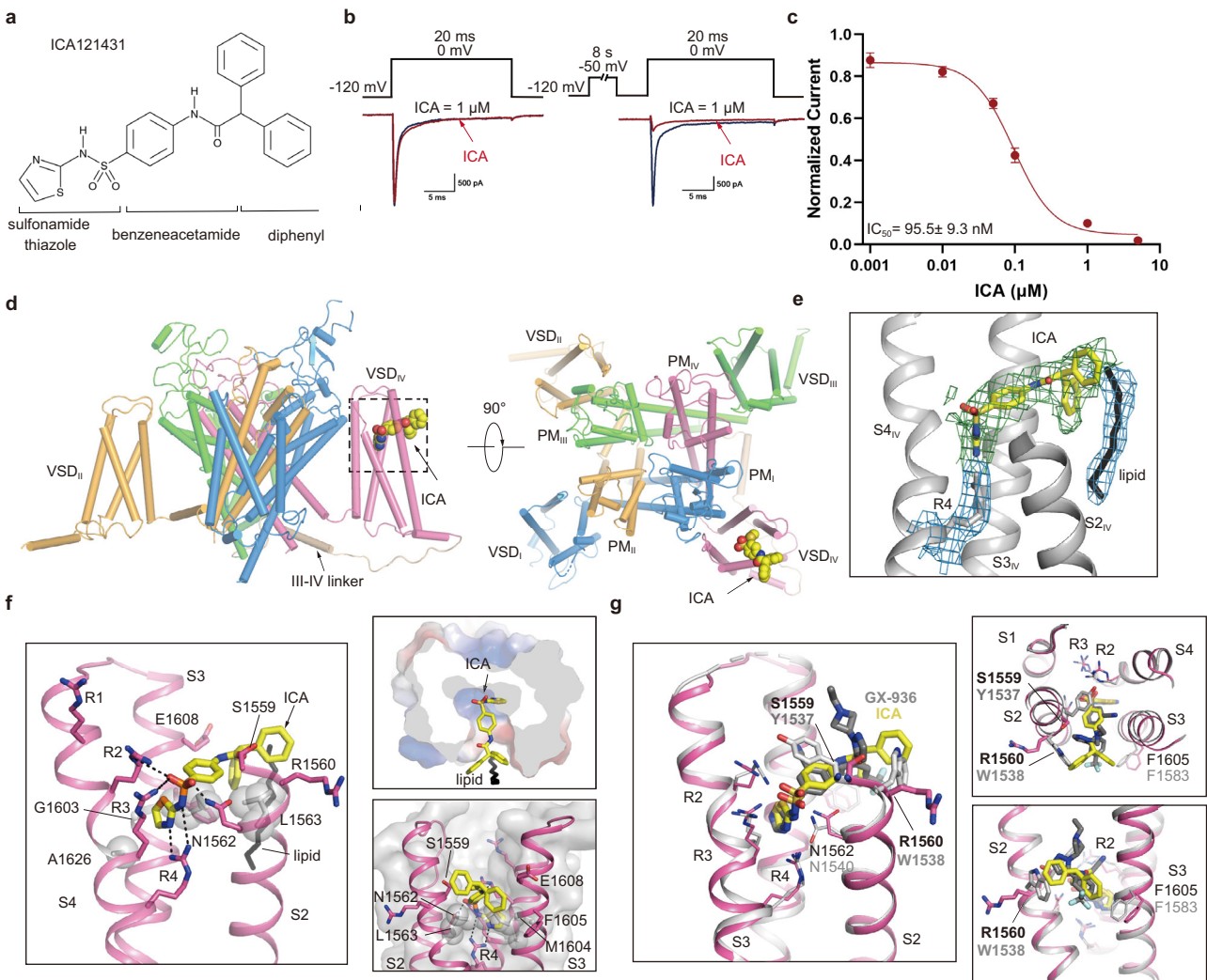

**Fig. 4 Inhibition of Na$_V$1.3 by ICA and the binding site for ICA. a** The chemical structure of ICA. **b** ICA preferentially inhibits Na$_V$1.3 at inactivated state. Current traces were recorded on Na$_V$1.3-transfected HEK293T cells, showing the effect of 1 μM ICA (red line) on Na$_V$1.3 using a resting-state inhibition protocol containing a 20-ms voltage step to 0 mV from holding potential (HP) at −120 mV (left panel), or an inactivated-state inhibition protocol composed of an 8-sec pre-pulse conditioning step at −50 mV to inactivate roughly half channels and back to −120 mV for 10-ms followed by a 20-ms test pulse at 0 mV from HP at −120 mV (right panel). Similar data were acquired from 6 cells for each protocol. **c** Dose-response of ICA on human Na$_V$1.3 using the inactivated-state inhibition protocol as described in panel (**b**). The Boltzmann distribution was fitted to each data of the normalized remaining peak current under different ICA concentrations, to yield IC$_{50}$ at 95.5 ± 9.3 nM. Data are mean + /− SEM acquired from 3–6 cells. **d** ICA binding site in Na$_V$1.3. The complex structure of Na$_V$1.3 is shown in side view (left panel) and top-down view (right panel), with ICA shown in a sphere model. The black dashed square indicates the area to be shown in panel (**f**). **e** Cryo-EM densities at 2.4σ for ICA (green mesh), R4 and a potential lipid molecule (blue mesh). **f** Detailed binding site for ICA inside the VSD$_{IV}$ of Na$_V$1.3. Side chains of key residues interacting with ICA are shown in sticks. Black dashed lines indicate electrostatic interactions between ICA and VSD$_{IV}$. **g** Comparison of binding sites for ICA in Na$_V$1.3 and GX-936 in Na$_V$1.7 (gray). Key residues determining isoform selectivity are labeled in black for Na$_V$1.3 and light gray for Na$_V$1.7, respectively. Source data are provided as a Source Data file.

Despite high sequence identity between Na$_V$1.3 and Na$_V$1.7 (Supplementary Table 1, identity at 85.7% among the structures), ICA selectively inhibits Na$_V$1.3/Na$_V$1.1 while another aryl sulfonamide antagonist GX936 selectively inhibits Na$_V$1.7[38]. Overlaying VSD$_{IV}$ of our Na$_V$1.3-ICA with that of the chimeric Na$_V$Ab-Na$_V$1.7/ VSD$_{IV}$-GX936 revealed that the interactions between the gating charges R2−R4 and the conserved thiazole group are almost identical in the two structures (Fig. 4g), which suggests that the strong electrostatic interactions between R2−R4 and the delocalized negative-charge of aryl sulfonamide are the key determinants for the potency of the inhibitors. This is consistent with the dramatic potency loss for GX936 when applied to the R4A mutant[38]. However, the middle benzene ring of ICA would be too close to the bulky side chain of Y1537 in Na$_V$1.7 if replacing S1559 of Na$_V$1.3; similarly, the

di-phenyl tail of ICA would directly clash with W1538 in Na$_V$1.7, whereas R1560 in Na$_V$1.3 stabilizes the tail (Fig. 4g). Sequence alignment around the antagonists binding site reveals that S1559 and R1560 of Na$_V$1.3 is only conserved in Na$_V$1.1 among the nine isoforms (Supplementary Fig. 7h). Our structure of the Na$_V$1.3-ICA complex illustrates that, despite high overall structure similarity, S1559 and R1560 are the key determinants for the selective inhibition of ICA on Na$_V$1.3/Na$_V$1.1, which is supported by significant potency reduction of ICA on the S1559Y and R1560W mutants by 9-fold and 33-fold, respectively[37].

**ICA preferentially binds to activated VSD$_{IV}$.** Comparing the VSD$_{IV}$ of Na$_V$1.3-ICA with that of Na$_V$1.3-BLA showed that the

two VSD$_{IV}$ are almost identical, both exhibit activated conformation with four gating charges (R1−R4) above the hydrophobic constriction site (HCS) (Fig. 5a). Most of the key residues around the receptor site assume nearly identical conformations, R4 of Na$_V$1.3-ICA was slightly pulled upward by the anionic thiazole group. This observation suggests that the ICA binding stabilizes VSD$_{IV}$ in the activated conformation. In contrast, when VSD$_{IV}$ of Na$_V$1.3-ICA is aligned with the deactivated VSD$_{IV}$ of rNa$_V$1.5 bound an α-scorpion toxin LqhIII[20], many residues interacting with ICA stay at similar positions except for the gating charges (Fig. 5b). In particular, the two helical-turns downward shift of the S4 helix in the deactivated VSD$_{IV}$ would make R1 in close proximity to the anionic thiazole group of ICA. Although it is possible for the R1 to adjust its sidechain to avoid potential clash with the antagonist, lacking of interactions from R2 and R3 destabilizing the ICA binding at the deactivated state. Similarly, alignment between VSD$_{IV}$ of Na$_V$1.3-ICA and the resting state VSD of Na$_V$Ab also illustrates that binding of ICA in the resting state is unfavorable due to lacking of the essential interactions from the gating charges (Fig. 5c). Together, these structure comparisons elucidate that ICA preferentially binds to the activated conformation of VSD$_{IV}$.

**Mechanism for inhibition of Nav by aryl sulfonamide antagonist.** Fast inactivation is the hallmark property of eukaryotic Na$_V$ channels, which closes the channel within 5 msec[12]. Dozens of mutations targeting fast inactivation gate cause disease-related gain-of-function phenotypes. Mutagenesis studies and high-resolution mammalian Na$_V$ structures suggested that the IFM-motif of fast inactivation gate allosterically closes the activation gate by binding to a hydrophobic receptor site near S6$_{IV}$[16] (Fig. 5d). Notably, binding of the IFM-motif is controlled by activation of VSD$_{IV}$. In other words, stabilizing VSD$_{IV}$ in activated conformation induces the binding the IFM-motif to its

receptor site and thus inhibits channel opening, while deactivated VSD$_{IV}$ destabilizes or prohibits the IFM-motif binding and thus enhances channel opening (Fig. 5d). Indeed, in our Na$_V$1.3-ICA structure, ICA binds to and stabilizes the activated VSD$_{IV}$ through multiple interactions including the three gating charges (R2−R4) directly involved. As a result, the IFM-motif binds to its receptor site, and the activation gate becomes non-conductive for hydrated Na$^+$ (Figs. 1c and 5e). This mechanism is likely applicable to other aryl sulfonamide antagonists that inhibit Na$_V$s, because the conserved anionic headgroup engages the conserved gating charges of VSD$_{IV}$ which determines the inhibition potency (Fig. 4f, g). Meanwhile, the chemically varied regions of the antagonists interact with distinct nearby residues which confers the drug isoform selectivity (Fig. 4g). Natural toxins target Na$_V$s and modify channel function. The α-scorpion toxins enhance Na$_V$s opening by trapping VSD$_{IV}$ in deactivated state which destabilizes the binding of IFM-motif to the receptor site (Fig. 5f). The binding of ICA to the deactivated or resting VSD is unfavorable, as a result, ICA exhibits poor inhibition of the channel in resting state (Fig. 4b).

## Discussion

Na$_V$1.3 is widely expressed in the central nervous system and is important for neuronal development. Many pathogenic mutations target Na$_V$1.3 and cause neurological disorders. The nine isoforms of Na$_V$s in human share high amino acid sequence and structure similarity. The central cavity of Na$_V$s harbors receptor sites for local anesthetic/anti-arrhythmic drugs and site-2 neurotoxins. However, the nearly identical chemical environment of the pore makes it less capable for isoform-selective drugs. Here we revealed two high-resolution cryo-EM structures of human Na$_V$1.3/β1/β2 in complex with a non-selective site-2 neurotoxin BLA as well as an isoform-selective antagonist ICA. BLA binds to the highly conserved central cavity, and the receptor site for BLA

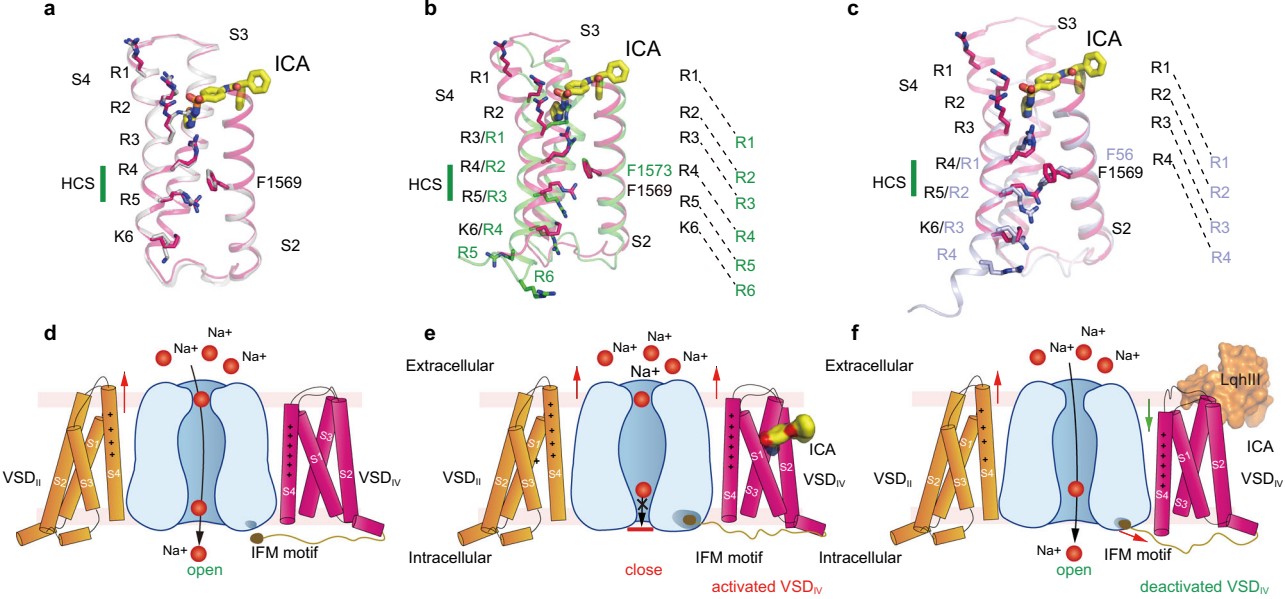

**Fig. 5 Allosteric inhibition mechanism for modulation of Na$_V$1.3 by ICA. a–c** Comparison of ICA-bound VSD$_{IV}$ of Na$_V$1.3 with activated apo-VSD$_{IV}$ of Na$_V$1.3 (**a**), deactivated LqhIII-bound VSD$_{IV}$ of Na$_V$1.5 (**b**), and the resting state VSD of Na$_V$Ab (**c**). ICA and side chains of gating charges shown in sticks. Green bar indicates the position of hydrophobic constriction site (HCS). Dashed lines indicate gate charge shifts at different states. **d** Activation of VSD$_{IV}$ controls fast inactivation of Na$_V$s. VSD$_{II}$ activation (red arrow) leads to gate opening, and VSD$_{IV}$ is not activated so the IFM motif is released from the receptor site. Domain I and III are hided for clarity. **e** ICA stabilizes the activated conformation of VSD$_{IV}$ of Na$_V$1.3 and induces inactivation of the channel. VSD$_{IV}$ is activated and stabilized by ICA, IFM motif binds tightly to the receptor site and closes the gate. **f** LqhIII stabilizes the deactivated conformation of VSD$_{IV}$ of Na$_V$1.5 and induces the opening of the channel. LqhIII binding pushes downward shift (green arrow) of S4$_{IV}$ and destabilizes the binding of IFM motif to the receptor site (red arrow), the release of IFM motif leads to channel opening.

is tightly packed by the P-loops and S6 helices from *D*I and *D*II. BLA accesses to the receptor site likely through the open activation gate, and a constricted gate at the inactivation state or closed gate at resting state surely prevents the entering of the bulky BLA. This structural observation could explain why the site-2 neurotoxins are in favor of binding to open state $Na_V$s. Once bound, BLA stabilizes the open conformation of S6 helices from both *D*I and *D*II, and prevents them shifting back to closed conformation. Therefore, BLA can activate $Na_V$s. Meanwhile, the bulky BLA also blocks the ion path leading to reduced peak current. Interestingly, other site-2 neurotoxins such as veratridine and batrachotoxin are more potent in activating $Na_V$s. This is probably because they share overlapping site for activating $Na_V$s, but causing different blockade based on their varied chemical structures. BLA has been used in clinic for the treatment of chronic pain and rheumatoid arthritis for almost four decades in China. Unfortunately, lack of isoform selectivity causes side-effects including cardiac arrhythmia and hyperexcitability, which limits its therapeutic applications. Therefore, finding and verifying new drug binding sites outside the pore is urgently needed for possible selective drug candidates which could help to reduce the off-target effects.

Activation of $VSD_{IV}$ is well-documented to induce fast inactivation of $Na_V$ channels. It has been shown that natural α-scorpion toxins target and trap $VSD_{IV}$ in deactivated conformation to prolong channel opening[20,51]. In contrast, stabilizing $VSD_{IV}$ in activated conformation can extend channel inactivation and therefore inhibit over-excitation. A group of aryl sulfonamide derivates showed promising $Na_V$s inhibition at nanomolar potency, and more importantly with isoform selectivity. For instance, ICA121431 was reported selectively inhibiting $Na_V1.3$ and $Na_V1.1$, while PF-04856264 and GX-936 inhibits $Na_V1.7$[37]. The mechanisms for ICA recognition and selective inhibition of $Na_V1.3$ are illustrated by our cryo-EM structure of $Na_V1.3/β1/β2$-ICA. The structure revealed that the antagonist only binds in the extracellular aqueous cleft of the activated $VSD_{IV}$ but not of the other three VSDs. The anionic headgroup of the antagonist engages the three gating charges on the S4 through electrostatic as well as van der Waals interactions. These interactions trap $VSD_{IV}$ in its activated conformation and thus promote channel inactivation. Therefore, the conserved anionic headgroup determines the potency of the aryl sulfonamide derivates. Meanwhile, the remaining part of ICA is recognized by surrounding sidechains to form a selective receptor pocket, in which S1559 and R1560 are key determinants for the isoform selectivity. Similarly, the varied region of GX-936 fits better within the $VSD_{IV}$ of $Na_V1.7$, and consequently it selectively inhibits $Na_V1.7$ over other isoforms. Comparison of the activated $VSD_{IV}$ of $Na_V1.3$-ICA and the deactivated $VSD_{IV}$ of $Na_V1.5$-LqhIII reveals that ICA indeed preferentially binds to activated $VSD_{IV}$ rather than deactivated or resting conformation. Taken together, our results demonstrate distinct mechanisms for modulation of $Na_V1.3$ by a non-selective site-2 neurotoxin BLA and an isoform-selective antagonist ICA. The structures help to explain the mixed activation and inhibition of $Na_V1.3$ by BLA as well as the potent and selective inhibition of $Na_V1.3$ by ICA at atomic level. These results provide important structural insights into $Na_V$s activation and inactivation, drug blockade, and developing better drugs targeting $Na_V$s with isoform selectivity.

## Methods

**Expression and purification of human $Na_V1.3/β1/β2$ complex.** The genes of human $Na_V1.3$ alternative splicing variant 2 (Missing Val625-Glu673, UniProt accession: Q9NY46-2), β1 (Uniprot accession: Q07699) and β2 (Uniprot accession: O60939) were amplified from HEK 293 cDNA library by PCR, which were further subcloned into a modified pEG BacMam vector (Supplementary Table. 2). To monitor protein expression and affinity purification, a mCherry fluorescent protein and a Twin-Strep tag were fused at the C-terminus of $Na_V1.3$. These constructs were confirmed by DNA sequencing. Recombinant baculoviruses were produced in Sf9 insect cells using the Bac-to-Bac baculovirus expression system (Invitrogen, USA). The Sf9 cells were cultured in ESF 921 medium (Expression Systems, USA) at 26 °C and 100 rpm. HEK293F cells were used to express recombinant protein, and were grown at 37 °C in the presence of 1% (v/v) fetal bovine serum in an incubator supplied with 5% $CO_2$ and shaking at 125 rpm. 1% (v/v) P2 baculoviruses of $Na_V1.3$, β1 and β2 were added to the medium when cell density reached $2 \times 10^6$ cells/ml. After 8−12 h, sodium butyrate was added to the culture at a final concentration of 10 mM and cells were incubated for another 48 h before harvesting.

The HEK293F cells expressing $Na_V1.3$-β1-β2 complex were harvested and resuspended in buffer A (20 mM HEPES pH 7.5, 150 mM NaCl, 2 mM β-mercaptoethanol (β-ME), and protease inhibitor cocktail including 1 mM phenylmethyl-sulfonyl fluoride (PMSF), 0.8 μM pepstatin, 2 μM leupeptin, 2 μM aprotinin and 1 mM benzamidine). Then the cells were broken by a Dounce homogenizer and the membrane fraction was enriched by ultra-centrifugation at $100,000 \times g$ for 1 h. Subsequently, the membrane proteins were extracted in buffer B (buffer A supplemented with 1% (w/v) n-Dodecyl-β-D-maltoside (DDM), 0.15% (w/v) cholesteryl hemisuccinate (CHS), 5 mM $MgCl_2$ and 5 mM ATP) for 1.5 h at 4 °C with rotation. Insoluble membrane fraction was removed by centrifugation at $100,000 \times g$ for 1 h. Then the $Na_V1.3$-β1-β2 complex was purified using Streptactin Beads, which was pre-equilibrated with buffer C (buffer A supplemented with 5 mM $MgCl_2$, 5 mM ATP and 0.06% (w/v) glyco-diosgenin (GDN)). Subsequently, the beads were washed with 10 column volumes of buffer C and buffer D (buffer C without 5 mM $MgCl_2$ and 5 mM ATP). The protein complex was eluted by 5 ml buffer E (buffer D plus 5 mM desthiobiotin). For $Na_V1.3$-β1-β2-ICA12131 complex, the eluted protein sample was supplemented with 50 μM ICA121431 and concentrated using a 100-kDa cut-off concentrator (Merck Millipore, Germany). The further purification was carried out by size exclusion chromatography (Superose 6 Increase 10/300 GL, GE Healthcare, USA) pre-equilibrated in buffer F (20 mM HEPES pH 7.5, 150 mM NaCl, 2 mM β-ME, 0.007% GDN, 50 μM ICA121431). Peak fractions were collected and concentrated to 4 mg/mL. Before cryo-EM sample preparation, the ICA121431 was added to the concentrated sample at a final concentration of 500 μM.

For $Na_V1.3$-β1-β2-BLA complex, the protein complex was purified similarly as the method described above. Bulleyaconitine A was supplemented throughout the whole purification process except SEC buffer. The final concentration of BLA was at 50 μM for buffer A, B, C and D, 100 μM for buffer E, and 200 μM for buffer F.

**Cryo-EM sample preparation and data collection.** Purified sample (2.5 μL) was applied to glow-discharged holey cooper grids (Quantifoil, 300 mesh, R1.2/1.3), and then was blotted for 2.0−5.5 s at 100% humidity and 4 °C before being plunged frozen in liquid ethane cooled by liquid nitrogen using a FEI Mark IV Vitrobot. All data were acquired using a Titan Krios transmission electron microscope operated at 300 kV, a Gatan K2 Summit direct detector and Gatan Quantum GIF energy filter with a slit width of 20 eV. All movie stacks were automatically collected using SerialEM at a calibrated magnification of 105,000× with a physical pixel size of 1.04 Å (super-resolution mode). Defocus range was set between −1.2 and −2.2 μm. The dose rate was adjusted to 10 counts/pixel/s, 2424 and 2858 movie stacks were collected for $Na_V1.3$-β1-β2-ICA and $Na_V1.3$-β1-β2-BLA, respectively, each stack was exposed for 6.4 s with 32 frames with a total dose of 60 e⁻/ Å².

**Data processing.** All the movie stacks were motion-corrected, binned by 2-fold and dose-weighted using MotionCor2[52], yielding a pixel size of 1.04 Å. Defocus values of each summed micrographs were estimated with Gctf[53]. Particle picking, 2D classification, 3D classification, polishing and CTF refinement was performed in RELION3.0[54]. A detailed data processing diagram was presented in Supplementary Fig. 2 and 4. The best class containing 175,513 particles for $Na_V1.3$-β1-β2-ICA and 152,431 particles for $Na_V1.3$-β1-β2-BLA were subjected to cisTEM[55] and cryoSPARC[56] for final refinement, respectively.

**Model building.** The structures of human $Na_V1.2$ (PDB code: 6J8E) alpha subunit and $Na_V1.7$ (PDB code: 6J8H) beta subunits were fitted into the cryo-EM density map of $Na_V1.3$-ICA and $Na_V1.3$-BLA using Chimera[57]. The models were manually checked and corrected in COOT[58] and subsequently refined in Phenix[59]. The model vs map FSC curves were calculated by Phenix.mtrage. Statistics for cryo-EM data collection and model refinement are summarized in Supplementary Table 3.

**Whole-cell voltage-clamp recordings of $Na_V1.3$ in HEK 293 T Cells.** HEK 293 T cells were cultured with Dulbecco's Modified Eagle Medium (DMEM) (Gibco) supplemented with 10% (v/v) fetal bovine serum (FBS) at 37 °C with 5% $CO_2$. HEK 293 T cells were transfected with two recombinant baculoviruses containing genes for $Na_V1.3$-mcherry and β1-eGFP/β2 (the same viruses were used for structural studies) for 12 h, then exchanged fresh DMEM supplemented with 10% FBS and 5 mM sodium butyrate culturing at 30 °C with 5% $CO_2$ for 1−2 day before recording. Electrophysiological experiments were performed to record $Na^+$ current

from the Na$_V$1.3 transfected HEK293T cells at room temperature (22−25 °C). Coverslips were placed in the recording chamber within bath solution containing 138 mM NaCl, 10 mM Hepes, 10 mM glucose, 5.4 mM KCl, 2 mM CaCl$_2$ and 1 mM MgCl$_2$ (pH 7.4 adjusted with NaOH). Pipette (2−3 MΩ) were prepared from borosilicate micropipettes by a Sutter P-97 puller and heat-polished before employment. The pipette solution contained 135 mM CsCl, 10 mM Hepes, 10 mM EGTA, 5 mM NaCl and 2 mM MgCl$_2$ (pH 7.4 adjusted with NaOH). Whole-cell voltage clamp recordings were obtained using EPC-10 amplifier (HEKA, Germany), with series resistances <5 MΩ and compensated by 60%. Drugs were diluted with the bath solution to the tested concentrations before recording and applied to the patched cell from pipette electrode with the diameter around 1 μm by puffing around via Picospritzer III microinjector. Data, at the sampling frequency of 20 kHz, were acquired through PatchMaster (HEKA) software and analyzed with Igor 6.2 (WaveMetrics).

All figures were prepared with PyMOL (Schrödinger, LLC) and Prism 8.0.1 (GraphPad Software).

**Reporting summary**. Further information on research design is available in the Nature Research Reporting Summary linked to this article.

## Data availability

The data that support the findings of this study are available from the corresponding author upon reasonable request. Atomic coordinates and corresponding EM maps of the NaV1.3-β1-β2-ICA PDB 7W7F and EMD-32343, and NaV1.3-β1-β2-BLA PDB 7W77 and EMD-32341 have been deposited in the Protein Data Bank (http://www.rcsb.org) and the Electron Microscopy Data Bank (https://www.ebi.ac.uk/pdbe/emdb/), respectively. The data that support the findings of this study are available within the article and its Supplementary Information. Source data of Fig. 1a, Fig. 4c and Supplementary Figs. 1a–b and 7a are provided with this paper. Source data are provided with this paper.

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

## Acknowledgements

We thank X. Huang, B. Zhu, X. Li, L. Chen, and other staff members at the Center for Biological Imaging (CBI), Core Facilities for Protein Science at the Institute of Biophysics, Chinese Academy of Science (IBP, CAS) for the support in cryo-EM data collection. We thank Yunlong Qiu, Yue Li, and Zhuoya Yu for their helpful discussions, Yan Wu for his research assistant service. This work is funded by Institute of Physics, Chinese Academy of Sciences (E0VK101 to D.J.), Chinese Academy of Sciences Strategic Priority Research Program (Grant XDB37030304 to Y.Z. and Grant XDB37030301 to X.C.Z.), the National Key Research and Development Program of China (Grant 2021YFA1301501 to Y.Z.), the National Natural Science Foundation of China (Grant 92157102 to Y.Z.), the National Natural Science Foundation of China (31971134 to X.C.Z.), the National Natural Science Foundation Regional Innovation and Development Joint Fund (U20A6005 to J.S.), and the Ministry of Science and Technology of the People´s Republic of China (2021ZD0203800 to S.Z.).

## Author contributions

D.J. and Y.Z. designed the experiments. X.L. and B.Y. made the constructs, expressed and purified protein complex sample. H.Z., Y.D., and X.L. prepared sample for cryo-EM study. Y.G. and X.L. collected cryo-EM data. D.J. processed the data, built and refined the models. H.X. and D.J. prepared figures. F.X. and Y.C.Z. collected the electrophysiology data. S.Z., X.C.Z., Y.Z., and D.J. analyzed and interpreted the results. J.S., X.C.Z., Y.Z., and D.J. wrote the paper, and all authors reviewed and revised the paper.

## Competing interests

The authors declare no competing interests.
