## [Peer Review File · Nature Communications]

Structural basis for modulation of human NaV1.3 by clinical drug and selective antagonistEditorial Note: This manuscript has been previously reviewed at another journal that is not operating a transparent peer review scheme. This document only contains reviewer comments and rebuttal letters for versions considered at *Nature Communications*.

REVIEWER COMMENTS

Reviewer #1 (Remarks to the Author):

The manuscript has been substantially improved during revision. I have no major questions. Two minor suggestions:

1. Fig4b: it would be nice to align the currents with the voltage- protocol above.
2. Fig3d: The author aligned BLA structure to NavAb structure and found the enlargement of gate in the presence of BLA. Is it possible that without BLA, the gate of Nav1.3 is larger than NavAb? Actually, in the new S6a1, the authors showed the gate in the presence of BLA or ICA are highly similar. Perhaps BLA could increase the probability of the gate in a more open conformation but this is not reflected in the current structure because the probability is still low. I would suggest a more conservative interpretation and careful structure comparison for the mechanism that BLA can enlarge the gate of Nav1.3.

Reviewer #2 (Remarks to the Author):

The thoughtful and thorough responses from the authors are well received and appreciated by this reviewer. I have no further comments.

Reviewer #4 (Remarks to the Author):

The manuscript by Li et al reports a cryoEM structure of Nav1.3 with the site 2 neurotoxin bulleyaconitine (BLA) and the VSD IV inhibitor ICA121431. While this is the first study to report the cryoEM structure of Nav1.3 (and any Nav structure with a site 2 toxin), it does not provide much additional insight to what is already known about the binding site and pharmacology of these two Nav modulators.

In Figure 3g, it is proposed that BLA does not bind to Nav1.3 in the resting state, however upon channel opening, BLA can enter the channel through the activation gate at the intracellular membrane surface. More experimental data is required to support this model. Is BLA really inactive at the resting state?

Craig et al (PMID: 31951114) demonstrated that the related site 2 toxin veratridine has different effects on Nav1.4 biophysics when pre-incubated at resting the state (holding potential –100 mV) or with 2 Hz pulses (–120 to 0 mV). Therefore, it would be important to include at a minimum the same control in this study for Figure 2b, comparing Nav1.3 current traces before and after incubation of BLA for 200 s at holding potential. To further ascertain if BLA can enter the channel through the activation gate on the intracellular membrane surface (as opposed to the fenestrations which would seem more likely as this is proposed for local anaesthetics) the authors could also compare the activity of BLA when applied in the intracellular solution or test activity with fenestration mutants – see El-Din et al - PMID: 30518562.

In Figure 2b and c, traces are only shown for n = 1 cell. The figure legend reports n = 6. Please include the n = 6 data in the form of a bar graph or current-time plot to ensure the traces shown are indeed

representative. Please provide a rationale for the protocol used in Figure 2c. It would have been more insightful to include a full IV. In Figure 2c, why does the control trace not return to baseline during the first pulse?

Minor:

Line 78 should be skin and not

shin Line 434 should be μm and

not μM

Response to Reviewers' Comments

Reviewer #1 (Remarks to the Author):

The manuscript has been substantially improved during revision. I have no major questions.

Reply: We thank Reviewer 1 again for his/her suggestions that helped to improve our manuscript.

Two minor suggestions:

1. Fig4b: it would be nice to align the currents with the voltage- protocol above.

Reply: We thank Reviewer 1's suggestion. The protocol and current trace have been aligned in the revision.

2. Fig3d: The author aligned BLA structure to NavAb structure and found the enlargement of gate in the presence of BLA. Is it possible that without BLA, the gate of Nav1.3 is larger than NavAb? Actually, in the new S6a1, the authors showed the gate in the presence of BLA or ICA are highly similar. Perhaps BLA could increase the probability of the gate in a more open conformation but this is not reflected in the current structure because the probability is still low. I would suggest a more conservative interpretation and careful structure comparison for the mechanism that BLA can enlarge the gate of Nav1.3.

Reply: We thank Reviewer 1's comment on this point. We agree with Reviewer 1 that the gate of the Nav1.3 in the presence of ICA or BLA are highly similar, this is because the two structures were determined at similar state. Although BLA does not enlarge the gate of Nav1.3 at inactivated state, we aligned Nav1.3-BLA with the resting-state sodium channel NavAb (PMID: 31353218, PDB code: 6P6W) in the Fig.3d and Fig.3e, showing that BLA binding prevents the S6 helices of DI and DII shift to resting state to closes the activation gate. Thus, BLA may activate Nav channels at more polarized membrane potential by stabilizing the S6 helices of DI and DII at open-state conformation.

Reviewer #2 (Remarks to the Author):

The thoughtful and thorough responses from the authors are well received and appreciated by this reviewer.

I have no further comments.

Reply: We appreciate Reviewer 2's positive comments and we thank Reviewer 2 again for his/her suggestions that helped to improve our manuscript.

Reviewer #4:

The manuscript by Li et al reports a cryoEM structure of Nav1.3 with the site 2 neurotoxin bulleyaconitine (BLA) and the VSD IV inhibitor ICA121431. While this is the first study to report the cryoEM structure of Nav1.3 (and any Nav structure with a site 2 toxin), it does not provide much additional insight to what is already known about the binding site and pharmacology of these two Nav modulators.

Reply: We appreciate Reviewer 4's positive comments and his/her suggestions for improving our manuscript. We report here the first cryo-EM structures of human Nav1.3 in complexes with a site-2 neurotoxin bulleyaconitine A (BLA) and Nav1.3-specific inhibitor ICA121431 (ICA). Previously reported electrophysiological and pharmacological studies of site-2 toxins and ICA121431 indeed provided important insights on the modulations of the small-molecule modulators on sodium channel, and point-mutagenesis studies revealed the site-2 toxins bind in the central cavity and ICA binds to the VSD4 of Navs. However, the central cavity harbors several receptor sites and how exactly BLA interacts with Navs remained elusive before this study. Our 3.3-Å high-quality EM structure of Nav1.3-BLA presents the first detailed interaction network for BLA binding, that is, BLA is sequestered in the cavity and DI-DII fenestration by multiple hydrophobic interaction and five hydrogen-bonds as illustrated in Fig. 2f. Importantly, three of the five hydrogen-bonds are formed between main-chain carboxyl group of T381, C941 and G942 are firstly identified in our structure (Fig. 2f). Besides these specific interactions, our Nav1.3-BLA structure provides the structural foundation to interpret the different effects of site-2 toxins on Nav channels, BLA binding stabilizes the open conformation of DI-S6 and DII-S6, and prevents them shift to closed resting-state, meanwhile, BLA blocks the ion path causing peak current reduction.

As for the Nav1.3 specific inhibitor ICA, Ahuja S. et al. (PMID: 26680203) reported the first structure of aryl sulfonamide derivative GX-936 bound to a 'non-functional chimeric' bacteria sodium channel, and explained the recognition and selective inhibition of Nav1.7-VSD4 by GX-936. However, the connection between selective antagonist binding and allosteric inhibition was lacking before this study. We employed the functional human WT Nav1.3 to achieve a full picture of antagonistic mechanism of the aryl sulfonamides, our structure clearly elucidates that ICA prefers binding to the activated conformation of VSD4 which couples to the fast inactivation gate and stabilizes the channel in inactivated state.

In Figure 3g, it is proposed that BLA does not bind to Nav1.3 in the resting state, however upon channel opening, BLA can enter the channel through the activation gate at the intracellular membrane surface. More experimental data is required to support this model. Is BLA really inactive at the resting state? Craig et al (PMID: 31951114) demonstrated that the related site 2 toxin veratridine has different effects on Nav1.4 biophysics when pre-incubated at resting the state (holding potential -100 mV) or with 2 Hz pulses (-120 to 0 mV). Therefore, it would be important to include at a minimum the same control in this study for Figure 2b, comparing Nav1.3 current traces

before and after incubation of BLA for 200 s at holding potential. To further ascertain if BLA can enter the channel through the activation gate on the intracellular membrane surface (as opposed to the fenestrations which would seem more likely as this is proposed for local anaesthetics) the authors could also compare the activity of BLA when applied in the intracellular solution or test activity with fenestration mutants – see El-Din et al - PMID: 30518562.

Reply: We thank Reviewer 4's suggestions on the resting-state and intracellularly applying BLA control experiments. As for the resting-state inhibition test, Nav1.3 expressing HEK293 cells were pre-incubated in the bath solution with or without 50 μ M BLA, current traces were recorded before (blue trace) and after (red trace) holding for 200 s at -100 mV, no inhibition was observed with 50 μ M BLA in the bath solution as shown in **new Supplementary Fig. 7b (right panel)** in the revision. In fact, under this 200 s holding potential at -100 mV protocol, increased current amplitude were observed by ~20% with (mean $122.2\% \pm 21.1\%$, n=5) or without (mean $119.7\% \pm 28.7\%$, n=4) incubation with BLA, suggesting that BLA is inactive at the resting state, which is consistent with previous report (PMID: 17585219). The observed increased current amplitude is likely due to recovery of almost all channels from inactivation after 200 s at -100 mV.

New Supplementary Fig. 7b BLA has no effect on Nav1.3 in resting state. Representative traces of Nav1.3 were recorded from holding for -200 s at -100 mV without or with 50 μ M BLA in the bath solution, respectively. A 20 ms test pulse at 0 mV was first recorded as a control trace (blue trace) from holding potential of -120 mV, then the patched cells were holding for 200 s at -100 mV, after 20 ms back to -120 mV, a second 20 ms test pulse at 0 mV was recorded (red trace) for either without or with 50 μ M BLA. Similar current traces were recorded from 5 cells with BLA and from 4 cells without BLA.

We also tested the effect of BLA when applied in the intracellular solution, as shown in **new Supplementary Fig. 7d**, BLA also progressively reduced peak current of Nav1.3. Interestingly, we found the inhibitory effect of BLA (mean $43.6\% \pm 0.8\%$, n=4) was weaker than that of extracellular application. Even though we started the 1,000 repetitive pulse protocol immediately after cell break, it would still take 3-5 min to finish the test. Since the BLA is known as a highly lipophilic molecule, we speculate that the tiny amount of BLA molecules in the 3 μ L

pipette solution may rapidly penetrate the plasma membrane and diffuse to the bath solution (2 mL) when we applied BLA in the pipette solution, and thereby decreased the concentration of BLA in the pipette solution. To test this possibility, we waited for 5 min after forming a giga-seal to allow the BLA diffusion and then recorded the current using the same protocol as shown in Supplementary Fig. 7d. It turned out the inhibition of BLA dropped to $19.9\% \pm 7.3\%$ ($n=5$), indicating that the intracellular BLA concentration may decrease (**See figure below, right panel**).

New Supplementary Fig. 7d Use-dependent inhibition of $\text{Na}_v1.3$ by BLA in intracellular solution. Representative current traces shown use-dependent inhibition of $\text{Na}_v1.3$ by $100 \mu\text{M}$ BLA in the intracellular solution. Current were elicited by

a 20 ms test pulse at 0 mV from holding

potential of -120 mV for 1000 repetitive pulses at 5 Hz. Similar current traces were recorded from 4 cells, yielding a mean inhibition of $43.6\% \pm 0.8\%$, $n=4$.

As we discussed in the manuscript, BLA only significantly affects $\text{Na}_v1.3$ at open-state in use-dependent manner, almost no effect at resting-state or inactive-state from our data and previous results (PMID: 17585219), we therefore hypothesis that BLA likely access the binding site through the open activation gate in the proposed model, but we did not rule out the possibility BLA may enter the binding site through the enlarged fenestration (**See below**).

- 235 Comparing to the open-state structure of $\text{Na}_v\text{Ab}^{49}$, the open activation gate appears to
236 be large enough for BLA to pass through (Fig. 3c), although one possibility cannot be
237 ruled out that BLA accesses the receptor site through enlarged fenestrations during state
238 transition (Fig. 3a). Once integrating into the receptor site formed by the X-crossing S6

In Figure 2b and c, traces are only shown for $n = 1$ cell. The figure legend reports $n = 6$. Please include the $n = 6$ data in the form of a bar graph or current-time plot to ensure the traces shown are indeed representative. Please provide a rationale for the protocol used in Figure 2c. It would have been more insightful to include a full IV. In Figure 2c, why does the control trace not to return to baseline during the first pulse?

Reply: We appreciate Reviewer 4's suggestions. We have revised the figure legends for Figure 2b and c in the revision. The representative current traces shown in Figure 2b and c were recorded from 1 cell respectively, similar results were recorded from 6 cells to confirm the effects of BLA. The current-time plot of data recorded from the 6 cells are shown in **new Supplementary Fig. 7a**.

New Supplementary Fig. 7a BLA inhibits Nav1.3 in use-dependent manner. Use-dependent block by BLA after 20-ms pulses at 5Hz from -120mV to 0mV (no BLA, squares; 50 μ M BLA, circles; data are mean \pm s.e.m.; n=6).

We also re-collected data for Figure 2c as suggested, the 1st test pulse was recorded for 20 ms rather than 10 ms in previous protocol. As shown in **new Figure 2c**, the control trace of the first pulse returns to baseline after 20 ms test pulse.

New Figure 2c BLA elicits weak activation of Nav1.3 after binding. Two-pulse protocol, which is composed of a first 20-ms test pulse at 0 mV, backing to -120 mV for 80-ms following a second 20-ms test pulse at -50 mV from HP at -120 mV, was used to elicit the activation of Nav1.3 by 50 μ M BLA for 1,000 repetitive pulses at 5-Hz frequency. Current trace of the first pulse (1P) and the 1,000 pulse (1000 P) are colored in black and purple, respectively. We achieved similar results from six different cells.

However, it has been consistently reported Nav1.3 current features a prominent slowly 'inactivating/persistent' current component of 0.5-15.4% of peak current when expressed in HEK293 cells (PMID: 1112239; 17381447; 20420834; 20675377). We also observed the persistent current component in some cases. We agree with Reviewer 4 that it would be more insightful to show the full I-V relationship curve. In this study, after 1000 repetitive pulses to reach the steady-state inhibition of Nav1.3 by BLA, the peak current was only 7.6% \pm 2.6% (n=4) of the first pulse. The tiny remaining current made it difficult to accurately calculate the shift of the voltage dependence of activation. Importantly, the residual BLA-induced current consistently appears but only yields a value of 3.3% \pm 0.6% (n=4) of the peak current of conditioning pulse 1 at potential of -50 mV that no current was observed without BLA. This result is consistent with previous report that the activation effect on sodium channel of BLA is weaker than aconitine and veratridine, and much weaker than grayanotoxin and batrachotoxin (PMID: 17585219; 925017).

Minor:

Line 78 should be skin and not shin

Reply: We have corrected it in line 80 in the revision.

Line 434 should be um and not uM

Reply: We have corrected it in line 449 in the revision.

REVIEWERS' COMMENTS

Reviewer #4 (Remarks to the Author):

The manuscript has been improved with the additional electrophysiology data. Interestingly, intracellular application of BLA was less effective when compared to extracellular application. The authors hypothesize that this is because BLA is highly lipophilic and is diffusing across the membrane and into the bath solution, thereby reducing the apparent concentration in the intracellular solution. However, it seems more likely that BLA has reduced efficacy because it does not diffuse across the membrane to the intracellular side to access the binding site via the open activation gate. And it is more likely that BLA accesses the binding site via fenestrations in a similar manner to local anesthetics, that are easier to access via extracellular application. While the authors do not exclude this possibility, the schematic in Figure 3g is not representative of this. Without further evidence that BLA does not access the binding site via fenestrations, Figure 3g should be amended or removed.

Response to Reviewers' Comments (Round 3)

Reviewer #4 (Remarks to the Author):

The manuscript has been improved with the additional electrophysiology data. Interestingly, intracellular application of BLA was less effective when compared to extracellular application. The authors hypothesize that this is because BLA is highly lipophilic and is diffusing across the membrane and into the bath solution, thereby reducing the apparent concentration in the intracellular solution. However, it seems more likely that BLA has reduced efficacy because it does not diffuse across the membrane to the intracellular side to access the binding site via the open activation gate. And it is more likely that BLA accesses the binding site via fenestrations in a similar manner to local anesthetics, that are easier to access via extracellular application. While the authors do not exclude this possibility, the schematic in Figure 3g is not representative of this. Without further evidence that BLA does not access the binding site via fenestrations, Figure 3g should be amended or removed.

Reply: We thank Reviewer 4 again for his/her suggestions that helped to improve our manuscript. Our electrophysiological results clearly showed BLA preferentially inhibits Nav1.3 at open state, two possible entrances for BLA accessing the binding site are opened gate and enlarged fenestration(s). Based on the size of the open gate and the strong use-dependent inhibitory effect of BLA, we proposed that BLA likely enters the cavity through the open gate, however, as we mentioned in the manuscript, we cannot exclude the possibility of fenestrations. With respect to Reviewer 4's comment about the possibility of fenestrations for BLA accessing its binding site, we decide to remove the model of Figure 3g, Figure 3a-f have presented all the information we discussed in the manuscript.